

# Downregulation of both mismatch repair and non-homologous end-joining pathways in hypoxic brain tumour cell lines

Sophie Cowman[1,2], Barry Pizer[3] and Violaine Sée[1]

[1] Institute of Systems, Molecular and Integrative Biology, Department of Molecular Physiology and Cell Signalling, University of Liverpool, Liverpool, Merseyside, United Kingdom
[2] Department of Pharmacology and Toxicology, College of Pharmacy, University of Utah, Salt-Lake-City, Utah, United States
[3] Paediatric Oncology, Alder Hey Children's NHS Foundation Trust, Liverpool, Merseyside, United Kingdom

Corresponding author
Violaine Sée,
violaine@liverpool.ac.uk

## ABSTRACT

Glioblastoma, a grade IV astrocytoma, has a poor survival rate in part due to ineffective treatment options available. These tumours are heterogeneous with areas of low oxygen levels, termed hypoxic regions. Many intra-cellular signalling pathways, including DNA repair, can be altered by hypoxia. Since DNA damage induction and subsequent activation of DNA repair mechanisms is the cornerstone of glioblastoma treatment, alterations to DNA repair mechanisms could have a direct influence on treatment success. Our aim was to elucidate the impact of chronic hypoxia on DNA repair gene expression in a range of glioblastoma cell lines. We adopted a NanoString transcriptomic approach to examine the expression of 180 DNA repair-related genes in four classical glioblastoma cell lines (U87-MG, U251-MG, D566-MG, T98G) exposed to 5 days of normoxia (21% $O_2$), moderate (1% $O_2$) or severe (0.1% $O_2$) hypoxia. We observed altered gene expression in several DNA repair pathways including homologous recombination repair, non-homologous end-joining and mismatch repair, with hypoxia primarily resulting in downregulation of gene expression. The extent of gene expression changes was dependent on hypoxic severity. Some, but not all, of these downregulations were directly under the control of HIF activity. For example, the downregulation of *LIG4*, a key component of non-homologous end-joining, was reversed upon inhibition of the hypoxia-inducible factor (HIF). In contrast, the downregulation of the mismatch repair gene, *PMS2*, was not affected by HIF inhibition. This suggests that numerous molecular mechanisms lead to hypoxia-induced reprogramming of the transcriptional landscape of DNA repair. Whilst the global impact of hypoxia on DNA repair gene expression is likely to lead to genomic instability, tumorigenesis and reduced sensitivity to anti-cancer treatment, treatment re-sensitising might require additional approaches to a simple HIF inhibition.

## INTRODUCTION

Glioblastoma (GBM) is a highly aggressive and infiltrative grade IV astrocytoma, the most common malignant brain tumour in adults (*Dolecek et al., 2012*). Survival rates remain low partly due to the limited treatment options available. Standard protocols involve initial resection of the tumour mass, which is challenging due to the undefinable tumour border. Therefore, post-surgery radiotherapy and chemotherapy are essential for disease control. Radiotherapy is typically delivered as multiple fractions with a 60 Gy total dose (*Chang et al., 1983*; *Omuro & DeAngelis, 2013*). Temozolomide an alkylating agent, is the sole chemotherapeutic agent used for GBM in the U.K. Using temozolomide as an adjuvant increases mean survival rate compared to radiotherapy alone (*Athanassiou et al., 2005*). However, in order for temozolomide to be effective, the $O^6$-methylguanine-DNA methyltransferase (MGMT) gene must be silenced, leaving non-silenced patients with even fewer treatment options (*Brada et al., 1999*; *Glassner et al., 1999*; *Esteller, Garcia-Foncillas & Andion, 2000*; *Hegi et al., 2005*). Alternative therapies currently under investigation include PARP inhibitors, and vaccine-based therapies (*Zhang et al., 2004*; *Cheng et al., 2005*; *Liau et al., 2018*). Despite many attempts to improve GBM outcome, survival over five years is close to zero, with numerous barriers to successful GBM treatment still remaining. One such barrier is tumour hypoxia, which can negatively influence the effectiveness of both radiotherapy and temozolomide treatment and is associated with poor patient survival for GBM and for many other solid tumours (*Marampon et al., 2014*; *Ge et al., 2018*).

Spatial and temporal heterogeneity of oxygen availability arises due to the formation of aberrant vasculature, prone to leaks and bursts, and large diffusion distances between oxygen rich vessels and tumour cells (*Vaupel, Rallinoâ & Okunieff, 1989*; *Vaupel & Harrison, 2004*). In the brain, healthy oxygen levels range from 5–8%, yet in GBM, cells are exposed to as little as 0.5% to 3% $O_2$ (*Rampling et al., 1994*), defined as pathophysiological hypoxia. In terms of reducing anti-cancer therapy effectiveness, hypoxia can alter DNA repair mechanism through epigenetic modifications, transcription and translation alterations, and post-translational modifications of DNA repair proteins (reviewed in *Scanlon & Glazer, 2015*). DNA damage response is the cornerstone of GBM therapy: both radiotherapy and temozolomide target DNA, causing high levels of DNA damage which overwhelms repair mechanisms leading to the induction of apoptosis via p53 activation (*Banin et al., 1998*; *Canman et al., 1998*; *Epstein et al., 2001*; *Turenne et al., 2001*; *Saito et al., 2002*).

Hypoxia has been shown to transcriptionally downregulate homologous recombination repair (HRR) components leading to reduced HRR capacity (*Bindra et al., 2004*, *2005*; *Meng et al., 2005*; *Bindra & Glazer, 2007*; *Bindra, Crosby & Glazer, 2007*; *Chan et al., 2008*). Also, components of nucleotide excision repair are downregulated by hypoxia even after periods of reoxygenation (*Dudás et al., 2014*). In addition, we previously showed that NBN and MRE11, members of a DNA double strand break recognition complex are downregulated by chronic but not acute hypoxia (*Cowman et al., 2019*). These studies

exemplify the fact that hypoxia can influence DNA repair, however, there has been little exploration of the global impact of long-term hypoxia on DNA repair gene expression in glioblastoma.

We adopted a NanoString transcriptomic approach to assess the impact of chronic hypoxia on DNA repair genes in glioblastoma cell lines. We found that hypoxia specifically affects mismatch repair (MMR), non-homologous end-joining (NHEJ) and homologous recombination repair (HRR). Additionally, we provide evidence that the hypoxia inducible factor (HIF) play a role in downregulation of some, but not all, key DNA repair genes. Downregulation of DNA repair genes by hypoxia will have significant clinical impact for cancer management and should be considered when designing new treatment methods and protocols.

# MATERIALS AND METHODS

## Reagents

Cell culture reagents were from Gibco Life Technologies and Foetal Calf Serum from Harlam Seralab (UK). Acriflavin was purchased from Sigma Aldrich. β-Actin (Ab8226), PMS2 (Ab110638) and anti-mouse HRP (Ab6808) antibodies were from abcam. HIF-1α (20960-1-AP) and HIF-2α (A700-003) were from Bethyl. Anti-rabbit HRP (7074S) antibody was from Cell Signalling.

## Cell culture

U87-MG (p53 wild type, MGMT methylated) and T98G (p53 mutant, MGMT unmethylated) were purchased from ATCC. U251-MG (p53 mutant, MGMT methylated) were purchased from CLS. D566-MG cells (p53 mutant, MGMT unknown) were a kind gift from Prof. DD Bigner (Duke University, USA). U87-MG and T98G were cultured in modified Eagle's Medium (EMEM) with 10% FCS, 1% Sodium pyruvate. D566-MG and U251-MG were cultured using EMEM, 10% FCS, 1% sodium pyruvate and 1% non-essential amino acids. Cells were maintained at 37 °C in 5% $CO_2$. For hypoxic experiments, cells were incubated in a Don Whitley H35 Hypoxystation for 1% $O_2$ or a New Brunswick Galaxy 48R incubator for 0.1% $O_2$. Cells were routinely tested for mycoplasma infection.

## NanoString assay

Cells were cultured in 6 cm dishes for 5 days at 21%, 1% or 0.1% $O_2$. RNA was extracted using High Pure RNA Extraction kit (Roche, Basel, Switzerland). A total of 100 ng of total RNA was used in the NanoString assay. The expression of 180 DNA repair genes, plus 12 housekeeping genes, were assessed using the NanoString nCounter Vantage™ RNA Panel for DNA Damage and Repair (LBL-10250-03) following the manufacturer's protocol. Analysis of the NanoString data was conducted using nSolver™ Analysis Software (3.0) with nCounter® Advanced Analysis plug-in (2.0.115). Data are available at https://www.ncbi.nlm.nih.gov/geo/query/acc.cgi?acc=GSE139250 and in Supplemental Dataset 1.

## Real time PCR

RNA was extracted using High Pure RNA Extraction kit (Roche, Basel, Switzerland). Reverse transcription was conducted using SuperScript® VILO (Invitrogen, Carlsbad, CA, USA) following the manufacturer's guidelines. RT-PCR experiments were performed and analysed as described in *Cowman et al. (2019)*. Primer sequences used were as follows: *LIG4* Forward: TCCCGTTTTTGACTCCCTGG Reverse: GGCAAGCTCCGTTACC TCTG, *ABL* Forward: TGGGGCATGTCCTTTCCATC Reverse: GATGTCGGCAGTGA CAGTGA, *ERCC4* Forward: CTCCCTCGCCGTGTAACAAA Reverse: ACACCAA GATGCCAGTAATTAAATC, *FEN1* Forward: GTTCCTGATTGCTGTTCGCC Reverse: ATGCGAATGGTGCGGTAGAA, *MSH5* Forward: GTTTGCGAAGGTGTTGCGAA Reverse: GTCTGAGACCTCCTTGCCAC, *PARP1* Forward: GCCCTAAAGGCT CAGAACGA Reverse: CTACTCGGTCCAAGATCGCC, *UBE2T* Forward: ATGTT AGCCACAGAGCCACC Reverse: ACCTAATATTTGAGCTCGCAGGT, *WRN* Forward: TCACGCTCATTGCTGTGGAT Reverse: CAACGATTGGAACCATTGGCA, *PMS2* Forward: AGCACTGCGGTAAAGGAGTT Reverse: CAACCTGAGTTAGGTCGGCA, *CYCLOA* Forward: GCTTTGGGTCCAGGAATGG Reverse: GTTGTCCACAGTCAG CAATGGT. Cyclophillin A was used as a housekeeping gene.

## Identification of hypoxia response elements

The DNA sequence for *LIG4* (NM_001352604.1) and *PMS2* (NM_000535.7) was obtained through the UCSC Genome Browser using human genome assembly h38 (*Kent et al., 2002*). The DNA sequence upstream of the first exon for each gene was examined for the consensus hypoxic response element, RCGTG, where R is A or G.

## Western blotting

Western blotting was performed as described in *Cowman et al. (2019)*. Briefly, 30–40 µg of protein were separated on 10% SDS–PAGE gels. Proteins were transferred onto nitrocellulose membrane (0.2 µm), followed by primary and secondary antibody incubation. Signal was developed using Amersham ECL Prime Western blotting Detection reagent (GE Healthcare, Chicago, IL, USA), and images taken using a G:BOX gel imaging system (Syngene, Cambridge, UK).

## Statistical Analyses

Student *T*-tests were performed using GraphPad Prism 6, for comparisons of means.

## RESULTS

### HRR, NHEJ and MMR are strongly regulated by hypoxia in a range of hypoxic GBM cell lines

To explore the global impact of tumour hypoxia on the expression of DNA repair genes in GBM cells, we used a NanoString assay. The NanoString nCounter gene expression system is a multiplexed assay, deemed more sensitive that RT-PCR and microarrays with less computationally heavy analysis than microarrays (*Geiss et al., 2008*). Four classical GBM cell lines (U87-MG, U251-MG, D566-MG and T98G), selected for their varied p53,
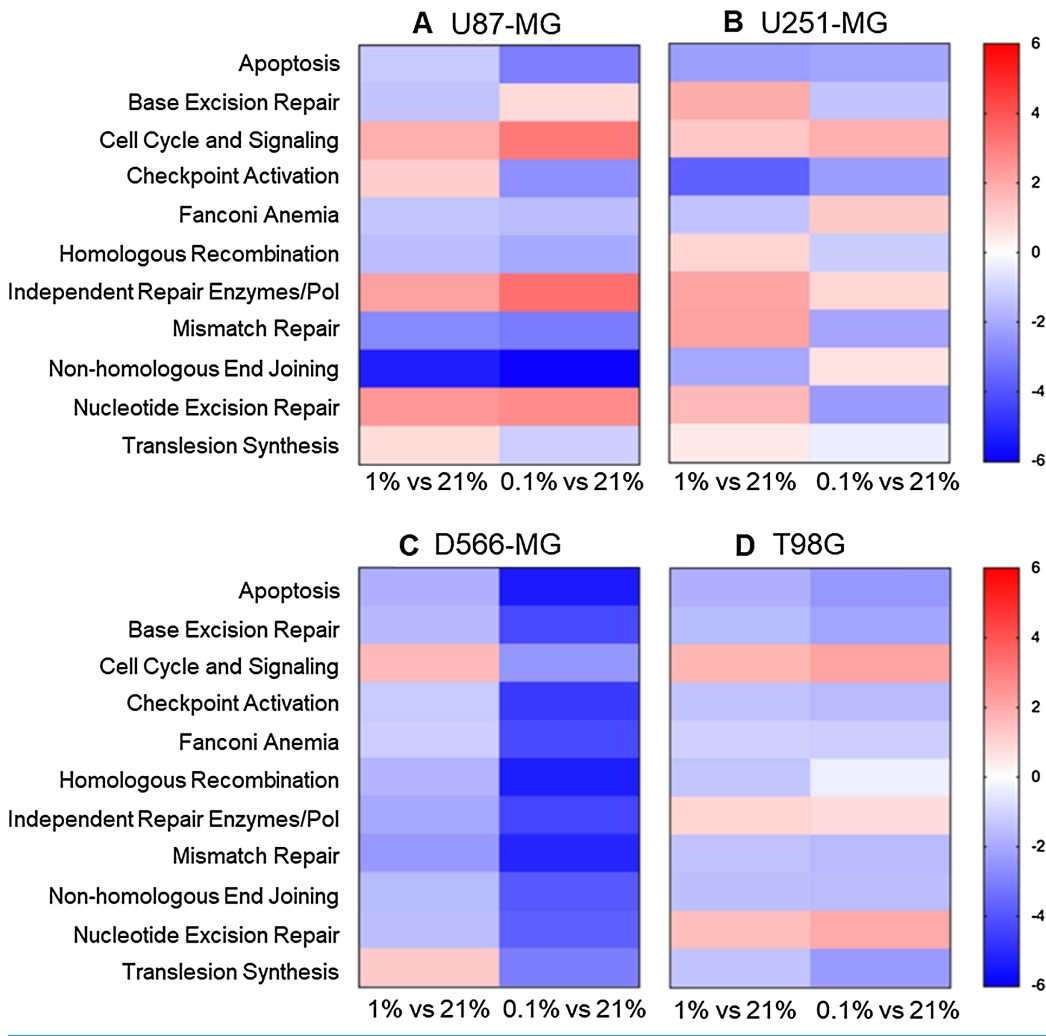

**Figure 1 Apoptosis, NHEJ, HRR and MMR are downregulated in multiple hypoxic GBM cell lines.** (A) U87-MG, (B) U251-MG, (C) D566-MG and (D) T98G cells were incubated in 21%, 1% or 0.1% $O_2$ for 5 days before gene expression analysis. Directed global significance scores were calculated for each defined annotation group and cell line, in 1% and 0.1% $O_2$, using the NanoString nSolver Advanced Analysis Software (see methods). Scores are represented in a heat map, where red and blue represent up and downregulation respectively. Data is representative of three independent experiments.

MGMT status, and hypoxia sensitivity were cultured in three different oxygen tensions (21%, 1% and 0.1% $O_2$), and *GLUT1* or *VEGFA* levels assessed by RT-PCR to confirm hypoxia had indeed resulted in hypoxia-induced gene expression changes (Fig. S1). Our previous work determined that hypoxia does not impact cell death, doubling time, or cell proliferation of these cell lines (*Richards et al., 2016*). The DNA repair genes assessed by the NanoString assay, were pre-assigned to annotation groups representing cell signalling and DNA repair pathways. Directed global significance scores (DGS), which measure overall differential regulation and direction, were calculated for each cell line in 1% and 0.1% $O_2$ (Fig. 1). Across all four cell lines, a number of pathways were downregulated, including apoptosis, mismatch repair (MMR), non-homologous

end-joining (NHEJ) and homologous recombination repair (HRR) (Fig. 1). MMR, NHEJ and HRR, are the core repair pathways involved in repairing temozolomide and radiation-induced DNA damage in GBM (*D'Atri et al., 1998*; *Cahill et al., 2007*; *Kondo et al., 2009*; *McFaline-Figueroa et al., 2015*). Whilst previous reports show conflicted results on hypoxia-induced changes to NHEJ, with both up and down-regulation of genes (*Scanlon & Glazer, 2015*), we here observed predominantly a downregulation of this pathway. In addition, the observed downregulation of the HRR pathway in hypoxia is in line with previous findings, and the impact of hypoxia on HRR efficiency has already been well characterised (*Bindra et al., 2004*, *2005*; *Meng et al., 2005*; *Bindra & Glazer, 2007*; *Bindra, Crosby & Glazer, 2007*; *Chan et al., 2008*).

Interestingly, increasing hypoxic severity from 1% to 0.1% $O_2$ drastically enhanced the regulation of each annotation group, especially in D566-MG cells (Fig. 1), and also increased the number of significantly regulated genes in this cell line (Fig. 2C). This correlates with the increase in *GLUT1* mRNA levels, a classical hypoxia marker, in 1% compared to 0.1% $O_2$ for D566-MG (Fig. S1). Beyond the specific D566-MG cells, more genes reached statistical significance at 0.1% $O_2$ in all cell lines, except for U251-MG (Fig. 2B), which had no dose-dependent response of *VEGFA* expression in hypoxia (Fig. S1). In U87-MG, 20 genes reached the strictest significance threshold of $p < 0.001$ in 0.1% $O_2$ compared to 10 genes in 1% $O_2$ (Fig. 2A). In contrast, T98G displayed strong fold changes yet few genes reached statistical significance (Fig. 2D). To validate the NanoString results and determine the robustness of the data, several gene candidates, which showed strong regulation in more than one cell line, were measured by RT-PCR and showed good consistency (Fig. S2). The NanoString assay therefore enabled us to successfully assess the extent of DNA repair gene expression alteration in GBM cell lines exposed to chronic hypoxia.

## Essential components of NHEJ and MMR are downregulated by hypoxia

The mismatch repair pathway removes incorrectly inserted nucleotides added during DNA replication, or removes modified bases generated by a DNA damaging agent. A sliding clamp composed of MutSα (MSH2, MSH6) and MutLα (MLH1, PMS2) actively translocates along the DNA in search of discontinuity (*Gradia, Acharya & Fishel, 1997*; *Blackwell et al., 1999*; *Gradia et al., 1999*; *Iaccarino et al., 2000*). EXO1 is responsible for nucleotide excision, and DNA polymerases replace excised nucleotides (*Longley, Pierce & Modrich, 1997*; *Zhang et al., 2005*). The MMR pathway is essential for effective temozolomide activity (*D'Atri et al., 1998*; *Cahill et al., 2007*; *McFaline-Figueroa et al., 2015*). The expression of 13 components of MMR were assessed in the NanoString assay (Figs. 3A and 3B). Whilst *MSH6*, part of the MutSα complex, was unaffected by hypoxia, *MSH2* was strongly downregulated 2.3-fold in D566-MG and 2.0-fold in T98G incubated in 0.1% $O_2$ (Fig. 3B), yet little change was observed at 1% $O_2$. *PMS2*, a central component of the MutLα complex and a key player in temozolomide resistance (*D'Atri et al., 1998*), was also downregulated in 1% $O_2$ and 0.1% $O_2$ for T98G (0.1% = 2.0-fold), D566-MG (0.1% = 2.6-fold) and U87-MG (0.1% = 1.6-fold) (Figs. 3A and 3B),

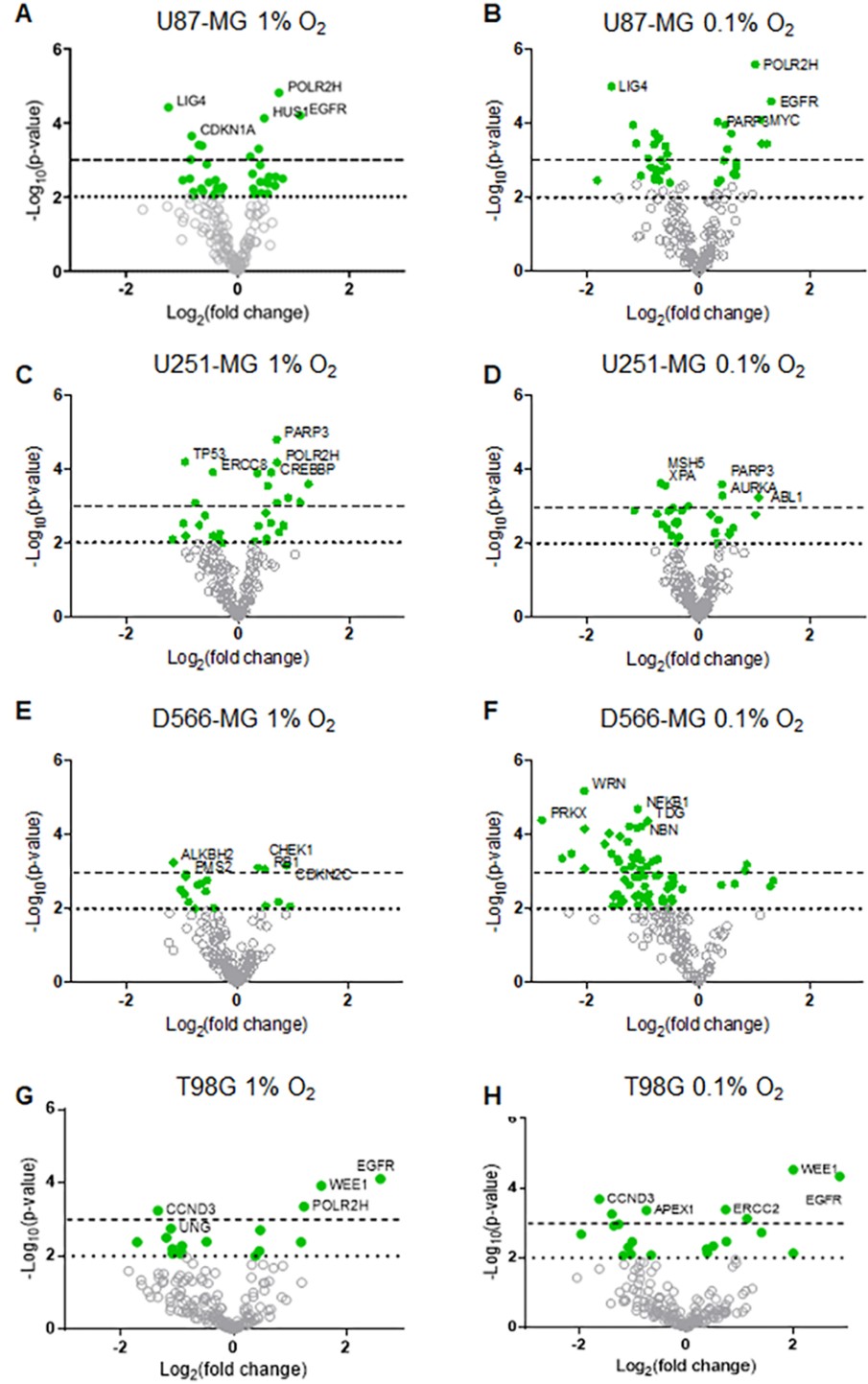

**Figure 2 DNA repair gene regulation by hypoxia is observed across all GBM cell lines.** Gene expression data for (A–B) U87-MG, (C–D) U251-MG, (E–F) D566-MG and (G–H) T98G in 1% or 0.1% $O_2$ is represented as $\log_2$ fold change with respect to 21% $O_2$ for each gene, plotted with $\log_{10}$ $p$-value. Each point represents the average of three experimental replicates. Green filled circles signify genes that have a $p$-value below the $p < 0.01$ threshold (dotted line). Dashed line is at $p = 0.001$. Grey hollow circles represent non-statistically significant genes. The top five statistically significant genes are identified with the gene name.

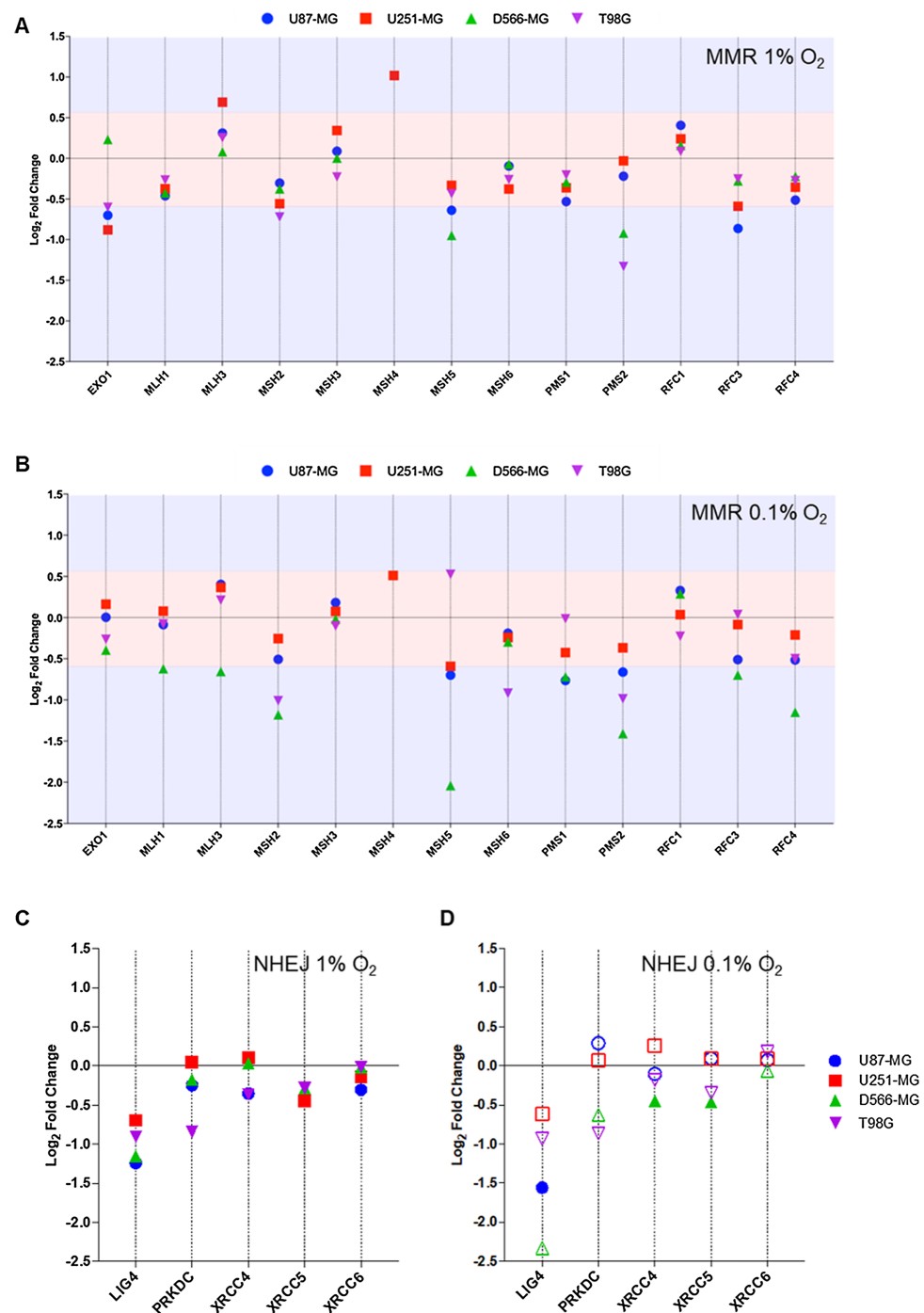

**Figure 3** *PMS2* and *LIG4*, essential components of MMR and NHEJ are downregulated in multiple cell lines. Differential gene expression was calculated for each gene as log₂ fold change for (A) MMR genes in 1% O₂, (B) MMR genes in 0.1% O₂, (C) NHEJ genes in 1% O₂, and (D) NHEJ in 0.1% O₂, with respect to the expression level at 21% O₂. Red shaded area denotes fold change values below the +/−0.58log₂fold (+/−1.5-fold) threshold of change. Blue area denotes fold change values above the designated threshold. Filled shapes represent data which is statistically significant ($p < 0.01$), whereas hollow shapes show no statistical significance. Data are represented as the mean of three independent experiments.

and was additionally a top hit in D566-MG (Fig. 2C). However, there was no change to *MLH1*, the binding partner of PMS2, in line with previous studies (*Koshiji et al., 2005*). Other changes include the downregulation of *MSH5* in a number of cell lines in 1% $O_2$ and 0.1% $O_2$, with particularly strong downregulation in D566-MG (4.1-fold) at 0.1% $O_2$ (Fig. 3B). MSH5 is primarily involved in meiosis (*Bocker et al., 1999*; *Snowden et al., 2004*), therefore the impact of these hypoxia-induced changes in brain tumour cell lines is debatable. However, activation of meiotic genes has been shown to aid in the initiation and maintenance of oncogenesis (*McFarlane & Wakeman, 2017*), potentially underlying a new unknown role of the hypoxia-induced changes in *MSH5*.

NHEJ is a highly error-prone process, yet is the primary pathway for DSB repair. A DNA-PK and Ku (Ku70/Ku80) complex recognises the DSB (*Gottlieb & Jackson, 1993*; *Kurimasa et al., 1999*). The ends of DNA are processed by various enzymes including Artemis, PNK, WRN and DNA polymerases (*Mahaney, Meek & Lees-Miller, 2009*), followed by re-joining by DNA Ligase IV complexed with XRCC4. Little consensus has been achieved as to the impact of hypoxia on NHEJ. Components of the damage recognition complex (Ku70-*XRCC6*, Ku80-*XRCC5*, DNA-PK-*PRKDC*) remain largely unaffected by hypoxia. However, in all cell lines tested, *LIG4* (DNA Ligase IV) was downregulated at least 1.5-fold in hypoxia (1% and 0.1% $O_2$.), although, this was only statistically significant in U87-MG (Figs. 3C and 3D). Interestingly, the extent of downregulation appears to strengthen with increasing hypoxia severity. In contrast, the expression level of *XRCC4*, the binding partner of DNA Ligase IV, was not impacted by hypoxia (Figs. 3C and 3D). Although, XRCC4 and DNA Ligase IV are both essential for effective NHEJ, there is no evidence to suggest their transcription is co-regulated. Therefore, it is feasible that hypoxia can impact the expression of each gene in different ways. Work by *Meng et al. (2005)*, observed similar hypoxia-induced gene expression changes for *LIG4*, with no change in *XRCC4* (*Meng et al., 2005*).

Overall, gene expression for components of both MMR and NHEJ were significantly impacted by hypoxia. Primarily, downregulation was observed, which could lead to reduced functionality of the repair pathways if the accessible pool of DNA repair proteins has been depleted. *PMS2* and *LIG4*, essential components of their respective repair pathways, are involved in chemoresistance (*D'Atri et al., 1998*; *Adachi et al., 2002*; *Kondo et al., 2010*), and were downregulated in at least two cell lines in both 1% and 0.1% $O_2$. Therefore, the mechanisms of regulation of these two genes was further examined.

## Chronic and acute hypoxic exposure leads to *PMS2* and *LIG4* downregulation

Despite the fact that hypoxic exposure in tumours is long-term, alterations to DNA repair gene expression by hypoxia are commonly explored upon acute hypoxic exposure. To determine the duration of hypoxia necessary to trigger downregulation of *PMS2* and *LIG4*, their level of expression in 1% $O_2$ was assessed over-time in D566-MG cells, the cell line in which the strongest regulation for these target genes was observed (Fig. 4A). Both *PMS2* and *LIG4* were downregulated after only one day of hypoxic exposure and remained downregulated over the 5-day period (Fig. 4A). Besides the transcriptional

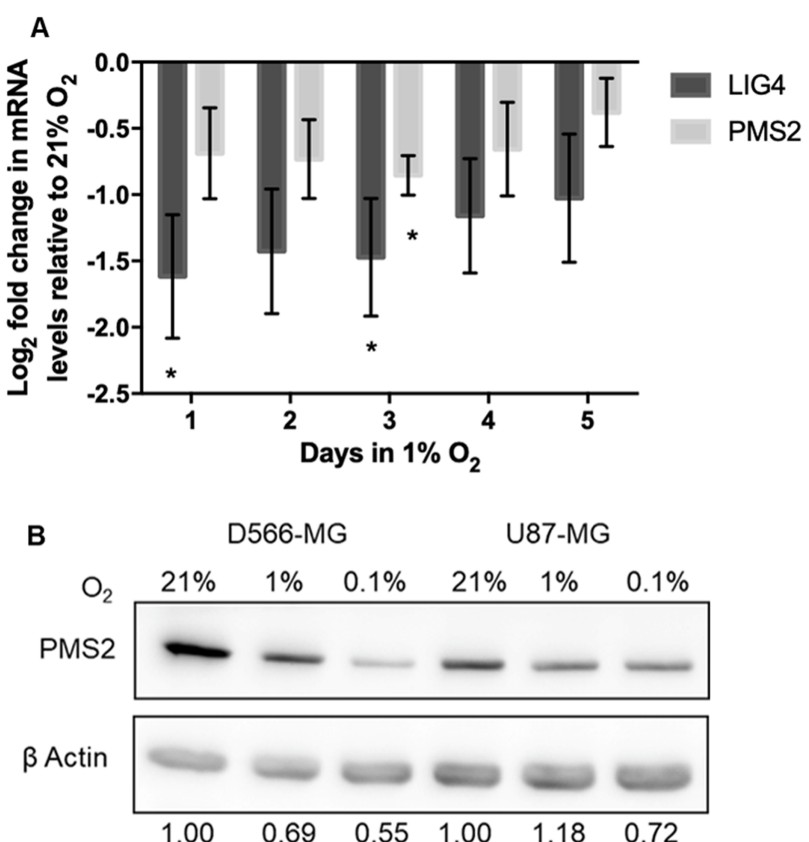

**Figure 4** **PMS2 and LIG4 are downregulated after acute and chronic hypoxic exposure.** (A) D566-MG cells were incubated in 1% $O_2$ for 1-5 days, as well as 21% $O_2$. After incubation, RNA was extracted and converted to cDNA for RT-PCR experiments, probing for *LIG4* and *PMS2*. Data is represented as $\log_2$ fold change in mRNA levels with respect to level for the 21% $O_2$ samples. Data are the mean of at least three independent experiments, with error bars showing S.E.M. * Denotes significant data where $p < 0.05$ indicated by one-way student *t*-test. (B) Western blot of PMS2 in D566-MG and U87-MG cells, incubated in 21%, 1% or 0.1% $O_2$ for 5 days. β actin was used as a loading control. Experiments were performed in triplicate, a representative blot is shown.

regulation, PMS2 protein levels were also downregulated by chronic hypoxia in D566-MG, and U87-MG (Fig. 4B), suggesting that the gene expression changes translate to the protein level. LIG4 protein levels could not be measured due to the lack of availability of a good quality antibody.

Previous studies demonstrated the involvement of HIF in the alteration of DNA repair gene expression. For example, downregulation of NBN, a component of HRR, arises due to the displacement of MYC by HIF at the NBN promoter (*To et al., 2006*). Examination of DNA sequence upstream of the initiating codon revealed that both *PMS2* and *LIG4* contained the consensus core hypoxia response element sequence RCGTG, where R is G or A. This suggests that HIF may be responsible for the hypoxia-induced regulation of *PMS2* and *LIG4*. The HIF pathway forms a negative feedback loop with the prolyl hydroxylase domain-containing protein 2 (PHD2) (*Stiehl et al., 2006*; *Bagnall et al., 2014*), resulting in a transient accumulation of HIF and lower levels after extended

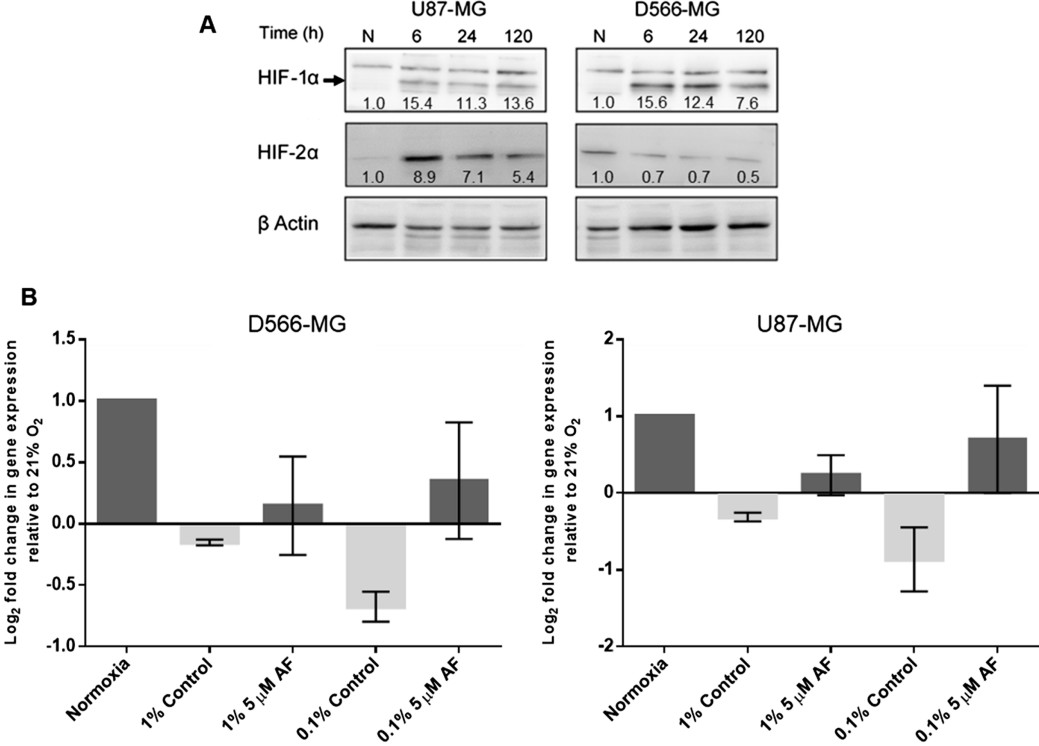

**Figure 5 HIF inhibition by Acriflavin restored *LIG4* downregulation.** (A) HIF-1α and HIF-2α are upregulated in chronic hypoxia. U87-MG and D566-MG cells were incubated in 1% O₂ and 21% O₂ for 0–120 h. Protein was extracted and the levels of HIF-1α and HIF-2α determined by western blotting in two independent experiments. A representative blot is shown with quantification of the bands, normalised to T0 (B) D566-MG and U87-MG cells were incubated in 21% and 1% O₂ for 24 h with and without 5 μM Acriflavin. *LIG4* mRNA levels were measured by RT-PCR. Data are represented as log₂ fold change in mRNA levels with respect to 21% O₂ samples. Data are the mean of at least three independent experiments, with error bars showing S.E.M.     

periods of hypoxia, typically after 12 h. However, in D566-MG and U87-MG cells, after 5 days of hypoxic exposure, HIF-1α remained high, with similar levels at 6 h and 24 h, and HIF-2α remained relatively consistent through the time course (Fig. 5A). To test the involvement of HIF in the expression levels of *LIG4* and *PMS2*, we initially aimed to use a silencing strategy. However, both D566-MG and U87-MG were highly sensitive to siRNA transfection performed to knockdown HIF, therefore a pharmacological approach was adopted. Acriflavin can inhibit HIF-1 and HIF-2 through binding to the PAS-B subdomain in the α subunit, preventing dimerisation of the α and β subunits, and reducing HIFs transcriptional activity (*Lee et al., 2009*). Treatment with acriflavin was sufficient to reduce hypoxia-induced *GLUT1* expression, a typical HIF target gene (Fig. S3A). In both D566-MG and U87-MG cell lines, acriflavin opposed hypoxia-induced *LIG4* mRNA regulation, although this was not statistically significant (Fig. 5B). In contrast, the level of *PMS2* was not altered by acriflavin treatment (Fig. S3B), suggesting different mechanisms of regulation for both genes, and that HIF is not the only player in hypoxia-induced regulation of DNA repair genes.

## DISCUSSION

Tumour hypoxia plays an essential role in tumour progression, metastasis and drug resistance, leading to poor patient outcome, particularly for GBM patients. Hypoxia has been shown to impact numerous signalling pathways including the cell cycle, metabolism and apoptosis, yet additionally, hypoxia can have a significant impact on DNA repair mechanisms. We have shown here that several key DNA repair pathways were downregulated by hypoxia, including MMR and NHEJ, both essential pathways for the repair of modified/mismatch bases and double strand breaks respectively. Further analysis determined that *PMS2* (MMR) and *LIG4* (NHEJ), essential components of their respective pathways, were downregulated across multiple cell lines after chronic and acute hypoxic exposure, providing potential targets for re-sensitisation of hypoxic tumour cells to DNA damaging therapies.

### Mismatch repair an essential repair pathway for temozolomide efficacy

We observed hypoxia-induced downregulation of *PMS2* (Figs. 3A and 3B), a vital component of MMR, involved in the search for discontinuity in DNA. MMR is required for repair of mismatched bases resulting from replication errors or DNA damaging agents. Defects in any of the critical MMR repair proteins (PMS2, MLH1, MSH2, MSH6), can lead to hypermutation and microsatellite instability, due to the increased number of point mutations (*Kim et al., 1999*). These point mutations drive tumorigenesis and cancer progression, which will directly contribute to poor patient outcome. In multicellular brain tumour spheroids, downregulation of *PMS2* and *MLH1* promoted the initiation of tumour cell formation and growth (*Collins et al., 2011*). In addition, in 1996, the link between MMR and cisplatin resistance was established, when cisplatin-resistant human ovarian adenocarcinoma cells were found to have reduced MLH1 protein (*Aebi et al., 1996*). In the same year, the downregulation of both MLH1 and MLH2 was found to induce cisplatin and carboplatin resistance (*Fink, Aebi & Howell, 1998*), although this was only a low-level resistance. We see little change in the gene expression of *MLH1* and *MLH2*, however, previously, we have reported that indeed, hypoxia causes low-level resistance to cisplatin in GBM cells (*Cowman et al., 2019*), which could be due to *PMS2* downregulation. MMR contribution to cisplatin toxicity has been well described, yet other DNA repair pathways may also play a role (*Pani et al., 2007*). More importantly for GBM, MMR is essential for temozolomide-induced apoptosis. The O6-methylguanine lesions inappropriately paired with thymine, are recognised and repaired by MMR. This results in thymine excision creating a long gap in the DNA which is filled and sealed. However, during this process thymine is matched again with the O6-methylguanine resulting in a futile repair cycle. This leads to the formation of a double-strand breaks during replication, which triggers cell death via apoptosis (*Klapacz et al., 2009*; *Karran & Bignami, 1994*; *Kaina & Christmann, 2019*; *D'Atri et al., 1998*). Thus, MMR is required for TMZ efficacy. Conversely, loss of double-strand break repair systems such as homologous recombination repair can enhance TMZ effectiveness as these newly formed breaks go unrepaired. Loss of the MMR component, MSH6, has been associated with increased tumour progression during temozolomide treatment in GBM (*Cahill et al., 2007*), and alterations

of *MSH2* expression can predict patient response to temozolomide therapy, with reduced expression correlating with decreased overall patient survival (*McFaline-Figueroa et al., 2015*). Downregulation of *PMS2* in hypoxia could have a significant clinical impact for GBM by contributing to increased mutation rate driving tumorigenesis and reduced sensitivity to temozolomide.

## DNA Ligase IV is crucial for NHEJ fidelity

NHEJ is essential for effective induction of apoptosis by temozolomide treatment (*Kondo et al., 2009*). Among the key components of NHEJ, we observed a significant hypoxia-induced downregulation of *LIG4* in multiple GBM cell lines, in line with previous observations (*Meng et al., 2005*). DNA Ligase IV is essential for re-joining broken DNA ends, yet is also required to prevent degradation of the ends of DNA, thus promoting accurate re-joining (*Smith et al., 2003*). Cell lines with hypermorphic mutations in *LIG4*, resulting in residual DNA Ligase IV function, are able to perform end-joining yet with reduced fidelity. In mice, loss of a single allele of *LIG4* results in the formation of soft tissue sarcomas, as a result of increased genomic instability (*Sharpless et al., 2001*). Therefore, downregulation of *LIG4* may reduce the effectiveness of NHEJ, thereby promoting genomic instability and further fuel tumorigenesis. However, in contrast, *LIG4* deficient cell lines have been shown to be more sensitive to ionising radiation (*Adachi et al., 2002*). Additionally, work by Kondo et al found that *LIG4* deficient cells were also more sensitive to temozolomide and Nimustine (ACNU), and siRNA of *LIG4* enhanced cell lethality of both chemotherapeutics (*Kondo et al., 2009*, *2010*). Although an increased genetic instability potentially arising due to *LIG4* downregulation may drive development and progression of GBM, the potential positive impact of increased temozolomide sensitivity may outweigh this negative implication. This highlights the double-edged sword of hypoxia, where both pro- and anti-cancer adaptations arise, which can be complex and difficult to untangle. Discovery of targeted therapies, which can exploit the anti-cancer components of tumour hypoxia would be advantageous, yet for this to occur further understanding of the molecular mechanism of hypoxia-induced changes needs to be gained.

## The role of HIF in hypoxia-induced DNA repair gene regulation

Long term hypoxia-induced changes in DNA repair gene expression can be orchestrated by a range of mechanisms, including epigenetic modifications. For example, alterations to histone methylation and acetylation at the promoters of *BRCA1* and *RAD51* leads to reduced gene expression (*Lu et al., 2011*). However, short-term transient changes to DNA repair gene expression are more commonly studied, with mechanisms both dependent and independent of HIF being described. Using an inhibitor of HIF, we showed that HIF is involved in the downregulation of *LIG4* but not *PMS2* in hypoxic GBM cells (Fig. 5B). HIF can directly modulate the transcription of genes through binding at the hypoxic response elements within promoters of DNA repair genes. However, the presence of hypoxic response elements does not guarantee direct HIF regulation. For example, in the *MLH1* promoter, hypoxia response elements have been identified (*Mihaylova et al., 2003*)

suggesting direct HIF regulation, yet a mechanism of MLH1 downregulation independent of HIF have also been discovered (*Mihaylova et al., 2003*; *Koshiji et al., 2005*; *Rodríguez-Jiménez et al., 2008*).

On the other hand, other genes are altered in a HIF-independent way, such as for example, via the E2F transcription factor binding in hypoxia. Indeed, in normoxia, the expression of *BRCA1* is mediated through the binding of the activating factor E2F1 as well as the E2F4/P130 suppressor. However, under hypoxic conditions, p130 binding to E2F4 is enhanced, due to alterations to p130 post-translational modifications. This results in an increase in E2F4/p130 transcriptional suppressor binding at the *BRAC1* promoter, reducing the rate of transcription (*Bindra et al., 2005*). Regulation of *RAD51* and *FANCD2* is thought to also occur by an E2F related mechanism (*Bindra & Glazer, 2007*; *Scanlon & Glazer, 2014*). The regulation of PMS2 may be through a similar mechanism.

## CONCLUSIONS

We have shown that chronic hypoxia results in the downregulation of multiple DNA repair pathways in GBM including essential components of MMR and NHEJ pathways. These alterations will likely not only impact chemotherapeutic treatment efficiency but also enhance the tumour genomic instability, hence further fuelling its development. The development of HIF inhibitors for cancer treatment is currently a popular area of research (reviewed in *Fallah & Rini, 2019*), and will undoubtedly be of great benefit for reducing the undesirable pro-tumorigenic impact of hypoxia. However, HIF is not the sole regulator of the hypoxia-induced reprogramming of the transcriptional landscape of DNA repair, as we have shown here with *PMS2* regulation. Whether HIF inhibition could be sufficient to re-sensitise hypoxic brain tumour cells to radio- and chemotherapy will require further investigation.

### Funding

This work was supported by the Alder Hey Oncology Fund. The funders had no role in study design, data collection and analysis, decision to publish, or preparation of the manuscript.

### Grant Disclosures

The following grant information was disclosed by the authors:
Alder Hey Oncology Fund.

### Competing Interests

Violaine Sée is an Academic Editor for PeerJ.

### Author Contributions

- Sophie Cowman conceived and designed the experiments, performed the experiments, analyzed the data, prepared figures and/or tables, authored or reviewed drafts of the paper, and approved the final draft.

- Barry Pizer conceived and designed the experiments, authored or reviewed drafts of the paper, provided clinical input, co-supervised the project and contributed to acquiring funding for the project, and approved the final draft.
- Violaine Sée conceived and designed the experiments, authored or reviewed drafts of the paper, and approved the final draft.

## Data Availability

Data is available at GEO GSE139250 and in the Supplemental Files.

## Supplemental Information

Supplemental information for this article can be found online at http://dx.doi.org/10.7717/peerj.11275#supplemental-information.

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
