# Peer review of "Downregulation of both mismatch repair and non-homologous end-joining pathways in hypoxic brain tumour cell lines"

_PeerJ, doi:10.7717/peerj.11275_

## Round 0.1 · original submission · Major Revisions

We appreciate your submission and hope you will take this opportunity to flesh it out with some of the experimental and presentation suggestions made by the reviewers. I would draw your attention particularly to Reviewer 1's suggestion in point 5, though points 1-4 and 9 should also be explicitly addressed, as well as Reviewer 2's four points. Please also update the introduction and discussion as requested. They agree, as do I, that the analyses of up- and down-regulated genes for all cells analyzed should be made easily accessible to the reader, if necessary as a supplementary table.

We understand that under the current circumstances, it is not necessarily easy to return to the laboratory to complete experiments and personnel is not always available, so if you need additional time beyond the suggested revision window, do not hesitate to ask.

Reviewer 1 ·

Basic reporting

In this manuscript by Crowman et al., the regulation of repair genes after hypoxia was examined. The results are not completely new, as similar studies have already been carried out on other tumor cell lines (e.g. Meng et al., 2005).

Critique
1) Four cell lines were used for the investigations. Why these lines? Do they have a different DNA repair or p53 status?
2) Although some repair genes are downregulated under hypoxic conditions, as Fig. 2 shows, there are different genes found to be downregulated in the cell lines. Thus ligase 4 is strongly downregulated in U87, but not in the other lines. Is there one or more genes that are downregulated in all 4 lines? How generalisable are the findings? How robust are the data (line 129) in terms of generalization. Is the downregulation of a specific gene related to the p53, IDH1 or MGMT status of the cells?
3) No data are provided on tumor material. Can the results be transferred to tumors?
4) Please insert a table in which the genes are clearly and transparently listed that are downregulated in the 4 lines. These cannot be seen in Figs. 1, 2 and 3.
5) MMR genes are cell cycle and proliferation dependently regulated. Thus, MSH2, which stabilizes MSH6, is positively regulated by E2F1. This transcription factor is down in non-proliferating cells. Therefore, it is very important to measure the proliferation level of the glioma cells after cultivation under hypoxic conditions. Are the effects observed a by-product of slowing down proliferation? Is the same observed if cells are cultivated under low serum condition? Please add data on cell's doubling time and cell cycle distribution under normal and hypoxic conditions.
6) line 216. The downregulation of MMR and DSB repair is discussed here and elsewhere in the mancript. Therapeutically, these repair pathways exert opposite effects. For example, a weakening of the TMZ effect is to be expected through downregulation of MMR, while an increase in the effect is to be expected after downregulation of HRR (as HRR is a major TMZ-protection mechanism). Possibly both effects neutralize each other. This should be discussed.
7) line 231: For cisplation and MMR quite old data are cited. Work published later by Jiricny and others showed that MMR is not clearly involced in the repair of cisplatin induced toxic adducts such as inter- and intrastrand crosslinks (for extensive discussion see Koeberle et al., Cisplatin resistance: preclinical findings and clinical implications, BBA, 1806, 172-182, 2010). This paragraph needs update.
8) line 238: MMR is essential for TMZ-induced apoptosis. Again, a quite old paper was cited. Please describe briefly why MMR is essential (there are many reviews on this, for a recent one see Kaina and Christmann, DNA repair in personalized brain cancer therapy with temozolomide and nitrosoureas, DNA Repair, 78. 128-141, 2019).
9) line 282: E2F1 regulates MSH2. Is downregulation following hypoxia mediated by upregulation of HIF and in turn downregulation of E2F1? I recommend to show in a western blot the expression level of E2F1, using the same cell extracts.
10) line 247: NHEJ, a highly error prone repair process,... This is not true, at least the general statement. Only the backup B-NHEJ pathway (also designated as Alt-NHEJ) was shown to be error-prone (XRCC1, PARP-1 and ligase III dependent), the major NHEJ pathway is largely error free.

Minor comment
11) line 10: Use the generally accepted terminology for repair genes, including MGMT. The full name is O6-methylguanine-DNA methyltransferase (MGMT).
line 111-113: This is a very superficial description of the function of MMR, NHEJ and HRR in temozolomide and radiation responses in GBM. At no point in the manuscript could I find a mechanistic description of MMR in TMZ-induced effects. Please describe in more detail. It is important to note that MMR does not remove TMZ-induced damage, but leads to futile repair cycles that activation death pathways.

Experimental design

see above

Validity of the findings

see above

Additional comments

see above

Reviewer 2 ·

Basic reporting

no comment

Experimental design

no comment

Validity of the findings

no comment

Additional comments

In this manuscript, authors studied the effect of chronic hypoxia on DNA damage repair gene expression in different glioblastoma (GBM) cell lines. They used the NanoString system to characterize the gene expression profile in cells treated with normoxia, moderate or severe hypoxia for up to 5 days. Several gene groups (apoptosis, NHEJ, HRR and MMR) were mostly found to be downregulated in four different GBM cell lines examined. Further, downregulation of PMS2 in MMR, and LIG4 in NHEJ were confirmed by RT-PCR and Western blotting. Finally, HIF inhibitor Acriflavin recovered the downregulation of LIG4, indicating that this process is mediated by the HIF-dependent mechanism.
The manuscript describes an important point on how cells respond to chronic hypoxia, and the molecular mechanism of gene downregulation under such condition. However, it is not clear how significant this machinery is upon GBM treatment. Following points should be addressed and modified before acceptance.

Points to be addressed
1. What would be the effect of Temozolomide on GBM cells cultured under chronic hypoxic condition compared with normoxic condition?
2. What would happen to the expression level of PMS2 and LIG4 upon reoxygenation?
3. Authors should show an entire expression profile of individual genes included in the Nanostring analysis.
4. In Figure 5, what is the expression level of LIG4 in normoxia? Statistical analysis should be done.

---

## Round 0.2 · Minor Revisions

There are only a few comments since the reviewers were satisfied overall with your previous thorough revision. Please try to respond to Reviewer 2's further comments to the best of your ability.

1. Line 234: add "oxygen" at the end of the sentence.

2. For the added sentence beginning "In both D566-MG and U87-MG cell lines, acriflavin abrogated hypoxia-induced LIG4 mRNA regulation... " I suggest that you prefer the term "seemed to counteract" or equivalent. You also should revise the legend to Figure 5 accordingly to be more reserved about the conclusions drawn, since indeed you did not establish a statistically significant effect either here or in the supplementary materials.

3. Copy editing: please remove the superfluous parentheses at line 79 "(reviewed in (Scanlon and Glazer, 2015))" and line 364 "(reviewed in (Fallah and Rini, 2019))"

I look forward to receiving this revision.

Reviewer 1 ·

Basic reporting

This is a revised version. A detailed criticism and comments were made in the primary review.

Experimental design

See above.

Validity of the findings

Significant findings were reported. The data are worth to be published.

Additional comments

The authors answered all of my questions and made satisfactory corrections in the manuscript.

Reviewer 2 ·

Basic reporting

N/A

Experimental design

N/A

Validity of the findings

N/A

Additional comments

The reviewer wonders if the chronic hypoxic treatment renders U-87MG cells sensitive to TMZ treatment due to changes in gene groups involved in DNA repair pathways.
It is not clear how the authors performed the experiment in point #1, but if it is without hypoxia, then they should examine the effect of chronic hypoxia.

---

## Round 0.3 · accepted · Accept

I very much appreciate your response and your adding in the final modifications requested, and am happy to accept your article for publication.